# The Cellular Structure and Mechanical Properties of Polypropylene/Nano-CaCO_3_/Ethylene-propylene-diene-monomer Composites Prepared by an In-Mold-Decoration/Microcellular-Injection-Molding Process

**DOI:** 10.3390/polym15173604

**Published:** 2023-08-30

**Authors:** Fankun Zeng, Xiaorui Liu, Yingxian Chen, Hao Li, Huajie Mao, Wei Guo

**Affiliations:** 1School of Automotive Engineering, Wuhan University of Technology, Wuhan 430070, China; zengfankun@whut.edu.cn; 2Hubei Key Laboratory of Advanced Technology for Automotive Components, Wuhan University of Technology, Wuhan 430070, China; lhstartrek@163.com (H.L.); maohj@whut.edu.cn (H.M.); 3Hubei Collaborative Innovation Centre for Automotive Components Technology, Wuhan University of Technology, Wuhan 430070, China; 4Guangqi Honda Automobile Research & Development Co., Ltd., Guangzhou 510700, China; chenyx@ghacrd.com.cn; 5School of Materials Science and Engineering, Wuhan University of Technology, Wuhan 430070, China; 6Institute of Advanced Materials and Manufacturing Technology, Wuhan University of Technology, Wuhan 430070, China

**Keywords:** polypropylene foams, cellular structure, toughness, microcellular injection molding, multiscale simulation

## Abstract

Polypropylene (PP)-composite foams were prepared by a combination process of microcellular injection molding (MIM) and in-mold decoration (IMD). The effect of ethylene propylene diene monomer (EPDM) on the crystallization properties, rheological properties, microstructure, and mechanical properties of PP-composite foams was studied. The effect of the additives on the strength and toughness of PP-composite foam as determined by the multiscale simulation method is discussed. The results showed that an appropriate amount of EPDM was beneficial to the cell growth and toughening of the PP blends. When the content of EPDM was 15 wt%, the PP-composite foams obtained the minimum cellular size, the maximum cellular density, and the best impact toughness. At the same time, the mesoscopic simulation shows that the stress concentration is the smallest, which indicates that 15 wt% EPDM has the best toughening effect in these composite materials.

## 1. Introduction

The development of lightweight automotives has increased the demand for foamed plastics. PP materials have been widely applied in the automobile industry [1,2,3]. However, the brittleness and poor cellular structure of foamed PP materials result in insufficient mechanical properties, and cannot meet the requirements of engineering applications. To broaden the application field, it is essential to improve the mechanical strength of foamed PP materials [4,5,6].

Reinforcements can effectively improve the mechanical strength and optimize the cell structure of foamed polymer materials; these materials include plastics, inorganic rigid particles, and elastomers [7]. Rachtanapun studied the cellular morphology and impact strength of the foamed high-density polyethylene/polypropylene (HDPE/PP) blends prepared by the batch foaming process. The result showed that the cell nucleation ability of the blends increased and the cellular morphology was satisfied when the composition ratio of HDPE/PP was, respectively, 50:50 and 30:70 [8]. Santiago studied the elastic phase and rigid phase of the bubble growth and mechanical strength of long-chain branched polypropylene (LCB-PP). The composites obtained a better cellular structure, improved ductility, and improved elastic modulus [9]. Mao studied the nano-CaCO_3_ relative to the foaming performance, crystallization, and strength of foamed-PP materials. It could be found that the crystallization of PP increased and the tensile strength of the composite foams reached their highest values by the addition of 6 wt% nano-CaCO_3_ [10]. Huang studied the foaming performance of HDPE/PP blends reinforced by nano-CaCO_3_. When the viscosity ratio of HDPE/PP blends was close to 1:1, the cell structure of the composite foams with 5 wt% nano-CaCO_3_ was the best [11]. Wang found that the talc could greatly promote the bubble nucleation and cellular structure of the PP/ talc blends. The results showed that the tensile toughness and impact toughness of PP/talc blends increased by 226% and 166%, respectively [12]. Zhao studied the effect of talc particle size on the foaming behavior and mechanical properties of PP/talc composite foams prepared by microcellular injection molding. The results showed that both micron and nanometer talc could effectively promote the crystallization and viscoelasticity of PP and improve cell density [13]. Li studied the toughening of thermoplastic polyurethanes (TPU)-reinforced PP foams and found that TPU increased the melt elasticity of PP. As the mass fraction of TPU increased, the impact strength of the blends first increased and then slightly decreased. A specific bimodal cell structure appeared when 5 wt% TPU was added [14]. Sancellan prepared nanocomposites by adding different mass fractions of Styrene-ethylene-butylene-styrene (SEBS) into PP and found that the impact toughness of the blends increased, while the tensile properties slightly decreased [15]. Maharsia studied foamed epoxy resin materials reinforced by rubber particles. The fracture strain of foamed epoxy resin material increased greatly with the addition of the rubber particles. The rubber particles absorbed much energy, significantly increasing the toughness of the foamed materials [16]. 

Inorganic particles and elastomers play different roles in foamed polymer materials [17,18]. Inorganic particles mainly improve the cell structure of the foamed materials by providing nucleation points. Cheng found that the cellular structure of foamed PP/EPDM materials could still be greatly improved by adding 4 wt% talc particles [19]. The elastomers mainly improve the impact strength by absorbing stress. According to the silver-shear band theory, elastomers, as the stress-concentration point, can induce a large number of silver stripes and shear bands and consume a lot of energy. Both elastomers and shear bands can prevent the silver pattern from expanding into cracks during the process of silver pattern expansion [20]. 

In this paper, EPDM and nano-CaCO_3_ were blended with PP and foamed by the IMD/MIM process. The effects of EPDM and nano-CaCO_3_ on the material properties, bubble growth, tensile strength, and impact strength of the PP-composite foams were studied. At present, the representative volume element (RVE) model, based on homogenization theory, has become one of the means to study the properties of macroscopic and microscopic materials. Wang established the RVE model of foamed polymethacrylimide (PMI) material to study its elastic properties. Comparing the predicted value with the experimental value, it was found that the RVE model determines the best prediction [21]. Malgorzata established the RVE model through the meso-mechanical method to study the strength and modulus of the foamed polyurethane carbon material. The analytical results show that the RVE prediction fits well with the experiment [22]. Therefore, the RVE model of PP-composite foams has been established, and the strengthening mechanism of additives and cells is discussed.

## 2. Materials and Experiments

### 2.1. Materials

The PP (K8303) was purchased from SINOPEC Yanshan Petrochemical Co., Ltd. (Beijing, China). The EPDM (3270P) elastomers were supplied by Dow Chemical Co., Ltd. (Midland, MI, USA). The nano-CaCO_3_ particles, with an average size of 60–80 nm, were provided by Zhejiang Changshan Jinxiong Co., Ltd. (Quzhou, China). The PP-G-MAH used to improve the compatibility of PP and additives was purchased from Dongyuan Ziheng Plastics Co., Ltd. (Heyuan, China). The commercial PET film, with a thickness of 0.2 mm, was selected as the interior decoration material. The nitrogen with 99% concentration was provided by Wuhan Xiangyun Industry and Trade Co., Ltd. (Wuhan, China). The commercial corrosive xylene was purchased from Sinopharm Chemical Reagent Co., Ltd. (Shanghai, China). Table 1 shows the materials’ properties.

### 2.2. IMD/MIM Process

The IMD/MIM progress is shown in Figure 1. The PP/nano-CaCO_3_/EPDM blends were heated and transferred to the injection molding machine chamber. The supercritical N_2_ was mixed with the molten blends. Then, the mixture was injected into the metal mold cavity decorated with a PET film. Thermodynamic instability occurred due to the decrease of pressure, leading to nucleation and growth of the bubbles [23]. The injection temperature section and mold temperature were set to 190 °C–200 °C–200 °C–190 °C and 25 °C, respectively. The cooling time was 16 s. Finally, we obtained the PP-composite foams after cooling and ejection. 

Six sets of samples were prepared. The first set was pure PP. The second set, called PP2, was PP + 6 wt% nano-CaCO_3_ + 5 wt% PP-g-MAH. The other sets were PP2 reinforced by 5 wt%, 10 wt%, 15 wt% and 20 wt% EPDM, respectively. The geometry of the sample is shown in Figure 2.

### 2.3. Material Characterization and Testing

#### 2.3.1. Rheological Behavior

The rheological properties of the samples were measured by a capillary rheometer (CR6000, Gotech, Dongguan, China). About 15 g of the samples were heated in a rheometer cylinder to 230 °C. The numerical data of viscosity and shear rate were recorded.

#### 2.3.2. Crystallization Behavior

About 10 mg of the samples were thermally analyzed in the N_2_ atmosphere using a thermal analyzer (DSC 214, Netzsch, Waldkraiburg, Germany). The samples were first heated from room temperature (about 25 °C) to 230 °C at a warming rate of 10 °C/min, held for 5 min, and cooled down to room temperature at a cooling rate of 20 °C/min to eliminate thermal history. The above cycle was repeated, and the data of the heating and cooling curves were recorded. The crystallinity is calculated by the following formula:(1)$$\chi_{c}\;=\;\frac{\Delta H}{\Delta H_{0}\;\times \;\phi }\;\times \;100\%$$
where Δ*H* and Δ*H*_0_ are the melting enthalpy of the PP composites and the pure PP with 100% crystallization, respectively. The value of Δ*H*_0_ is 209 J/g. The symbol *ϕ* is the mass fraction of PP.

#### 2.3.3. Microstructure

The microstructure of the cells and distribution of the additive particles were characterized by scanning electron microscopy using a JSM-IT300 (JSM-IT300, JEOL Ltd., Tokyo, Japan). The average cell size *D* (μm) and cell density *N* (cells/cm^3^) were calculated by the following formula:(2)$$D=\frac{\sum_{i=1}^{n}d_{i}}{n}$$
where $d_{i}$ is the diameter of a single cell, and *n* is the number of cells in the selected area.
(3)$$N={\left(\frac{n\times M^{2}}{A}\right)}^{\frac{3}{2}}$$
where *M* is the magnification in the selected area, and *A* is the area (cm^2^).

Figure 3 shows the illustration of the bubble growth in the IMD/MIM progress. When the melt flows into the mold cavity, the polymer melt will be subjected to different shear stress, resulting in the deformation of the bubbles. Length–diameter ratio (d) and deviation angle (*θ*) were used to reflect the degree of deformation of the bubble. The calculation formula of d is as follows:(4)$$d=\frac{b}{a}$$

#### 2.3.4. Mechanical Performance Test

An electronic universal mechanical testing machine (MST CMT6104, Shenzhen, China) was used to characterize the tensile strength of the samples. An Izod impact-testing machine (XJUD-5.5, Chengde, China) was used to test the impactor strength of the samples. All of the tests were conducted ten times at room temperature, and the averages of the data were taken as valid experimental results.

### 2.4. The Construction of RVE and Analysis Process

Both the addition of additives and the formation of a cellular structure have impacts on the mechanical properties of the composite foams. Based on the experimental data, the RVE model of composite foams was constructed to study the toughening mechanism of additives and cells, as shown in Figure 4. First, we tested the tensile stress–strain curves of each set of solid PP composites and entered them into the Digmat 2017 software as material data. Then the PP composites were homogenized, that was, the additives were considered to be uniformly distributed in the PP matrix. The cell size and density of the core layer and the transition layer were measured, respectively. The equivalent cell information (cell shape, cell size, and cell density) in the RVE model unit was determined according to experimental data of cell shape, average cell size, and expansion ratio. The size of the RVE model unit was 0.2 mm × 0.2 mm × 0.2 mm.

To analyze the effect of additives and cells on the mechanical strength of PP-composite foams, the stress-concentration factor (*α*) was introduced to characterize the stress distribution in the composite foams as follows:(5)$$\alpha =\frac{\sigma_{max}}{\sigma_{n}}$$
where $\sigma_{max}$ is the maximum equivalent stress, and $\sigma_{n}$ is the average equivalent stress.

To simplify the calculation, we have made the following assumptions:(1)PP composites were considered to suit an elastic–plastic model, which meant that the curves of tensile stress and strain conformed to Hooke’s law.(2)The core-layer bubbles were spherical, and the transition-layer bubbles were ellipsoid.(3)The uniaxial peak strain ε_11_ = 0.03 was used for model loading.

The stress-concentration factor was studied by adding load to the RVE unit.

## 3. Results and Discussion

### 3.1. Rheological Behavior

Figure 5 shows the effect of adding nano-CaCO_3_ and EPDM on the viscosity of PP composites. It can be seen that the viscosity of the composites improved by adding nano-CaCO_3_. The main reason lies in the role of “nailing” that nano-CaCO_3_ played in the melt, one which limited the relative movement of the PP molecular chain, increasing the viscosity. With the addition of EPDM, the viscosity of the composites at first decreased and then increased at a low-shear-rate level. According to the rheological principle, the addition of an excessive quantity of elastic particles will interfere with the shear flow field of the continuous phase of the matrix, causing the system to consume additional capacity, which is manifested as the increase of viscosity in the macroscopic scale. The melt viscosity has a great influence on the cell structure of the composites. If the viscosity were to be too high, the cell nucleation and growth would be inhibited, and the gas would be concentrated in this region, resulting in localized small cells and large cells in the composites. Furthermore, the increase in viscosity will hinder the uniform distribution of elastic particles in the melt, affecting the overall properties of the composites.

### 3.2. Thermal Analysis

Figure 6 shows the crystallization behavior of the PP-composite foams. Compared to the pure PP, the *T_c_*, *T_c_*_(*onset*)_, and $\chi_{c}$ of the PP phase in the composite foams all increased after adding nano-CaCO_3_ and EPDM. This occurs owing to the roles of heterogeneous nucleation that nano-CaCO_3_ and EPDM play in the matrix. The increases of *T_c_* and *T_c_*_(*onset*)_ indicate earlier crystallization in the forming process, which can effectively inhibit the deterioration of the cellular structure. It can be seen in the melting curves in Figure 6b that the addition of EPDM does not have much influence on *T_M_*, forming the crystal peak around 168 °C. Meanwhile, it is also established that, after the addition of EPDM, the melting peak of *β* crystal gradually disappears, which indicates that the addition of EPDM will inhibit the formation of anisotropic crystals. However, excessive elasticity will reduce the crystallinity of PP. As the EPDM increased from 10 wt% to 20 wt%, the $\chi_{c}$ decreased from 46.3% to 45%, as depicted in in Figure 7.

### 3.3. Distribution of Elastomer Particles

As is shown in Figure 8, EPDM presented a uniform “sea-island” distribution in the PP matrix. EPDM had good interface compatibility with PP after the addition of PP-g-MAH. As the content of EPDM increased, the particle size gradually decreased. This indicated that the dispersion of EPDM had increased. According to the silver-shear band theory [20], EPDM particles could then be the stress-concentration points in the matrix, causing a large number of silver stripes, which could hinder the development of the crazing into destructive cracks.

### 3.4. Microcellular Structure

In the IMD/MIM process, the shear stress perpendicular to the melt-flow direction was greater than that parallel to the melt-flow direction, which caused severe deformation of the cell. The bubbles in the core layer tended to be spherical, while those in the transition layer tended to be ellipsoidal. Therefore, the morphologies in both directions were studied.

#### 3.4.1. The Microcellular Structure Perpendicular to the Melt-Flow Direction

Table 2 shows the microcellular structure perpendicular to the melt-flow direction in the core layer and the transition layer. Comparing the microcellular structure of PP2 with PP, it is obvious that the cell density of the core layer and the transition layer have increased, and the cell size decreased. This indicates that nano-CaCO_3_ was beneficial in improving the foaming properties of PP. With the addition of EPDM, the cell density in the core layer did not change significantly, but the cell density in the transition layer increased significantly and the cell size decreased. However, when the content of EPDM increased to 20%, the cell of the core layer became larger.

Figure 9 shows that the cell size of the transition layer was smaller than that of the core layer, but the cell density was higher. This was mainly caused by the difference in cooling rate between the transition layer and the core layer. Compared with the core layer, the transition layer was closer to the mold, which had a faster cooling rate and shorter cooling time. This resulted in the inhibition of cell growth and consolidation. When 6 wt% nano-CaCO_3_ and 15 wt% EPDM were added to PP, the cellular structure became optimal, and there was no large cell in the core layer. At this time, the cell size was the smallest and the cell density was the largest. However, when the mass fraction of EPDM increased to 20 wt%, the cell diameter of PP-composite foams in both the core layer and the transition layer increased, while the cell density decreased. Although EPDM could act as the nucleating agent to promote nucleation, excessive EPDM (20 wt%) would tangle with PP molecules, reduce the crystallization of PP, and lead to the easier merger of bubbles. Furthermore, the relative mass fraction of PP decreased owing to the addition of 20 wt% EPDM, resulting in a decrease in the degree of cell nucleation and cell density, respectively.

#### 3.4.2. The Microcellular Structure Parallel to the Melt-Flow Direction

Table 3 shows the microcellular structure parallel to the melt-flow direction in the core layer and the transition layer. Due to the large shear force, the bubbles in the transition layer showed an obvious shear deformation. The offset angle and the length–diameter ratio of the transition layer were studied. As the EPDM content increased from 0 wt% to 20 wt%, the offset angle gradually decreased, the ratio of length–diameter increased, and the cells tended to be flat. When the viscosity increased, the migration resistance of the melt flow also increased, resulting in greater deformation of the bubbles and an increase in the length–diameter ratio [26,27].

#### 3.4.3. Density and Expansion Ratio

Figure 10 shows the density and expansion ratio of PP-composite foams. As the EPDM content increased from 0 wt% to 15 wt%, the expansion ratio of PP-composite foams increased from 123.4% to 134.8%, and the density decreased from 0.777 g/cm^3^ to 0.698 g/cm^3^. When the EPDM content increased to 20 wt%, the expansion ratio of the composite foams decreased significantly, and the density increased. According to the mixture rule, the addition of EPDM is conducive to the weight reduction of solid PP composites because of its lower density. However, excessive EPDM would lead to the deterioration of the cell structure of the PP-composite foams, causing a decrease of expansion ratio and an increase of density.

### 3.5. Mechanical Properties

Figure 11 shows the tensile properties of PP-composite foams. The pure PP had the lowest tensile elongation and strain-at-break. The tensile strength of PP was improved by adding nano-CaCO_3_ particles. The addition of rigid particles improved the rigidity of the PP matrix. Additionally, as a nucleating agent, nano-CaCO_3_ particles promoted the crystallization of PP and improved the cell structure. All of these factors were conducive to the improvement of the tensile strength of the PP foams.

As the EPDM content increased, the tensile stress and yield strength of the composite foams decreased, while the tensile elongation at first increased and then decreased. Since the modulus and strength of EPDM were both much lower than that of the PP, when the content of EPDM increased, the modulus and strength of PP composites would decrease. EPDM improved the tensile elongation of the PP-composite foams, but when EPDM was at 20%, the cell structure of composite foams deteriorated, resulting in a decrease in elongation.

Figure 12 shows the impact properties of PP-composite foams. The impact strength of PP2 increased after adding nano-CaCO_3_ particles, which could be attributed to the improvement of microcellular structure. After adding EPDM, the impact toughness of PP-composite foam was further improved. The EPDM acted as the stress-concentration points, causing more cracks and creating shear bands, resulting in the PP-composite foams absorbing more energy before generating an impact fracture. However, when the content of EPDM increased to 20 wt%, the impact toughness decreased. Previous studies have shown that cell structure deteriorates when EPDM content is 20 wt%. This proves that the toughness of the composite foam depends not only on the EPDM content but also on the cell structure.

### 3.6. Multi-Scale Simulation Analysis

Table 4 shows the equivalent stress distribution of the RVE model (Time = 1) under strain loading perpendicular to the melt-flow direction. Figure 13 shows the curves of loading time and stress-concentration factor of the section perpendicular to the melt-flow direction at different time steps, as calculated by Equation (5). The stress-concentration degree of the core layer was higher than that of the transition layer, which made it easier for destructive cracks to develop. As the EDPM content increased from 0 wt% to 15 wt%, the stress-concentration factor of the core layer decreased, and the stress-concentration factor of the transition layer increased. The stress-concentration factor of the transition layer of PP2 + 15 wt% EPDM was the lowest, while the impact strength reached its highest level. However, when EPDM content increased to 20 wt%, the change in stress-concentration factor reversed. At the same time, the cell structure deteriorated, and both tensile strength and impact strength decreased. This indicates that the cellular structure parameters had an important effect on the mechanical properties of PP-composite foams.

Table 5 shows the equivalent stress distribution of the RVE model (Time = 1) under strain loading parallel to the melt-flow direction. Figure 14 shows the curves of loading time and stress-concentration factor of the section perpendicular to the melt-flow direction at different time steps, as calculated by Equation (5). With the addition of nano-CaCO_3_ and EPDM, the stress-concentration factor of the transition layer changed greatly, while that of the core layer changed little. The stress-concentration degree of PP2 was the highest compared to the others. At the same time, the tensile tests indicated that PP2 had the highest tensile strength. This indicates that while nano-CaCO_3_ can increase the tensile strength and improve cell structure, it can also lead to the increase of stress-concentration points. After adding EPDM, the stress-concentration factor of the transition layer decreased significantly, indicating that EPDM could reduce the stress-concentration phenomenon. This was conducive to improving the toughness of the composite foams.

## 4. Conclusions

In the IMD/MIM process, the effects of additives on foaming behavior and mechanical properties of PP were studied. Both inorganic particles and elastomers could be used as heterogeneous nucleating agents to improve PP’s foaming properties. The tensile strength and impact strength of PP-composite foams were improved simultaneously by the addition of 6 wt% nano-CaCO_3_. However, with the increase of EPDM content, the tensile strength of PP foams decreased, and the impact strength at first increased and then decreased. When the EPDM content was 15 wt%, the PP-composite foams had the best cellular structure, while the impact strength reached its highest value, at 38.2 kJ/m^2^. At the same time, the stress-concentration factors in both the section perpendicular to the melt-flow direction and the section parallel to the melt-flow direction were at a low level, which made it easy to terminate the cracks. The cell structure and elastomers showed a synergistic toughening effect.

## Figures and Tables

**Figure 1 polymers-15-03604-f001:**
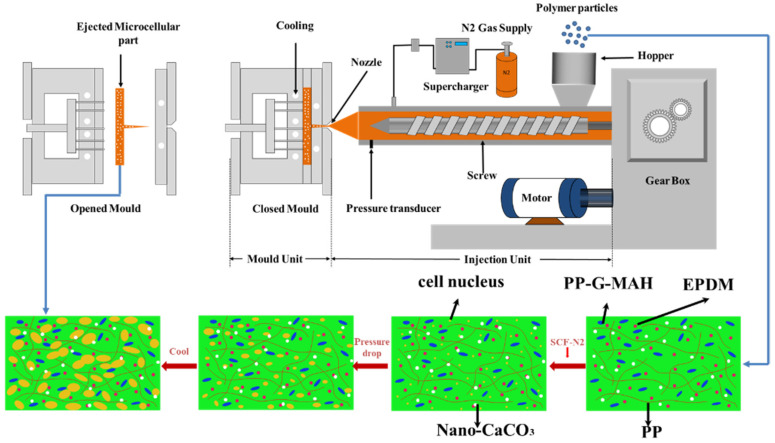
Illustration of IMD/MIM progress [24].

**Figure 2 polymers-15-03604-f002:**
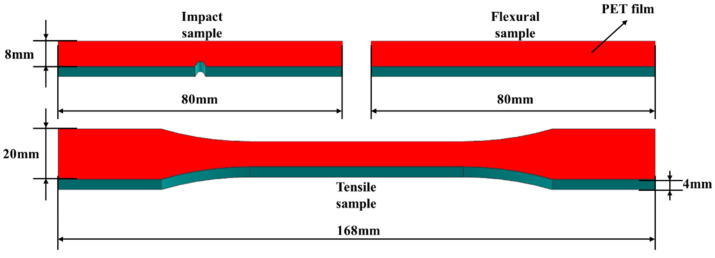
The geometry of the samples.

**Figure 3 polymers-15-03604-f003:**
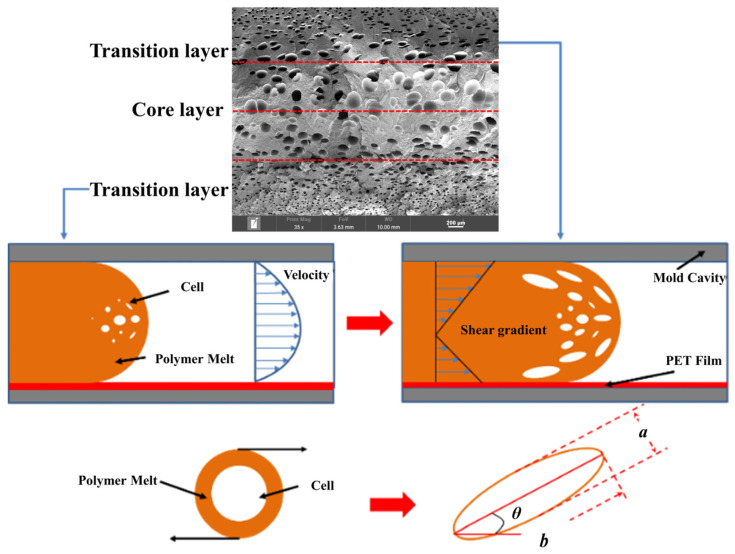
Illustration of the bubble growth in the IMD/MIM progress [24].

**Figure 4 polymers-15-03604-f004:**
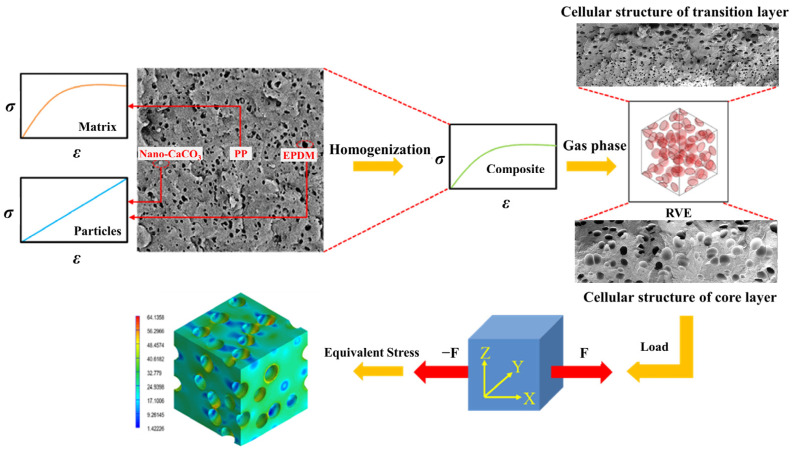
Illustration of the RVE model analysis process.

**Figure 5 polymers-15-03604-f005:**
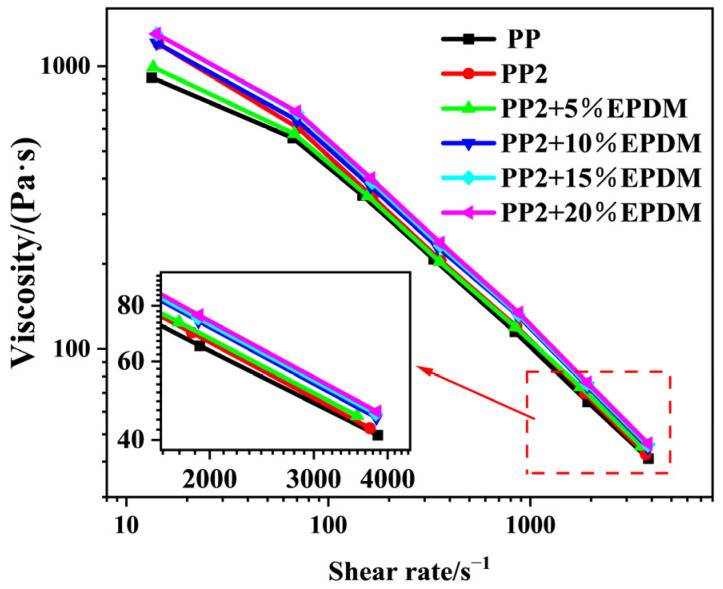
Shear-rate-viscosity curves of PP composites.

**Figure 6 polymers-15-03604-f006:**
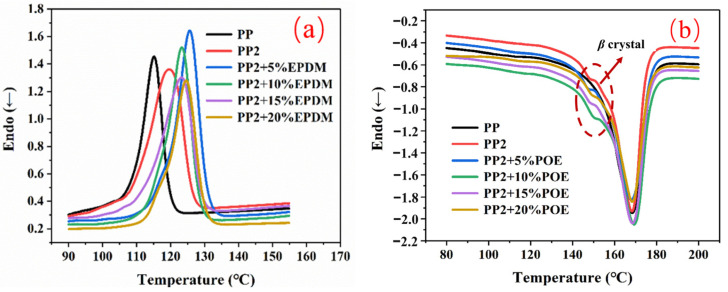
Crystallization behavior of PP-composite foams: (**a**) crystallization curves; (**b**) melting curves.

**Figure 7 polymers-15-03604-f007:**
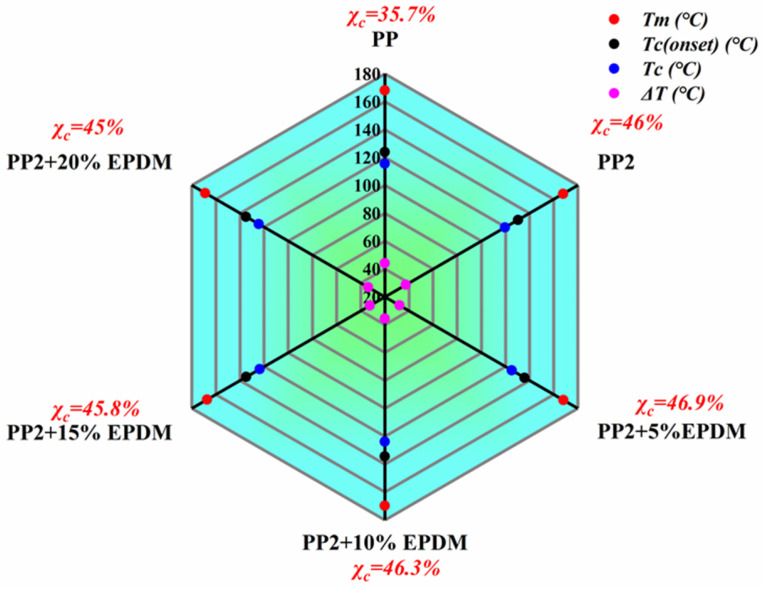
DSC thermal analysis parameters (*T_m_—*extreme point of melting peak; *T_c_*_(*onset*)_*—*initiation temperature of crystallization; *T_c_—*extreme point of crystallization peak; Δ*T—*Temperature difference between crystallization peak and melting peak; $\chi_{c}$*—*crystallinity).

**Figure 8 polymers-15-03604-f008:**
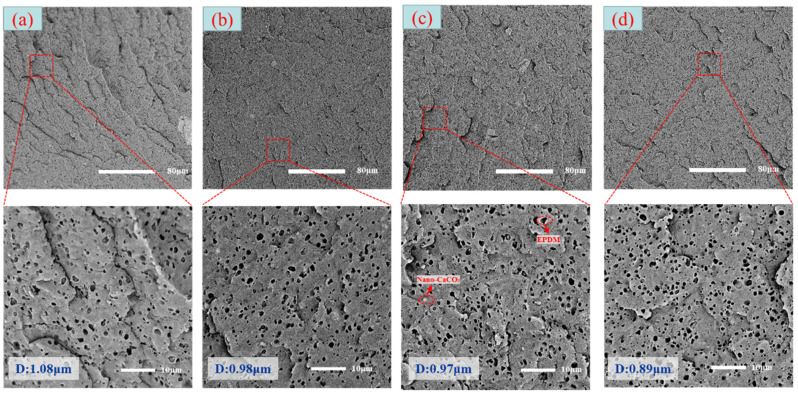
SEM image of EPDM un-foamed composite corrosion: (**a**) PP2 + 5 wt% EPDM; (**b**) PP2 + 10 wt% EPDM; (**c**) PP2 + 15 wt% EPDM; (**d**) PP2 + 20 wt% EPDM.

**Figure 9 polymers-15-03604-f009:**
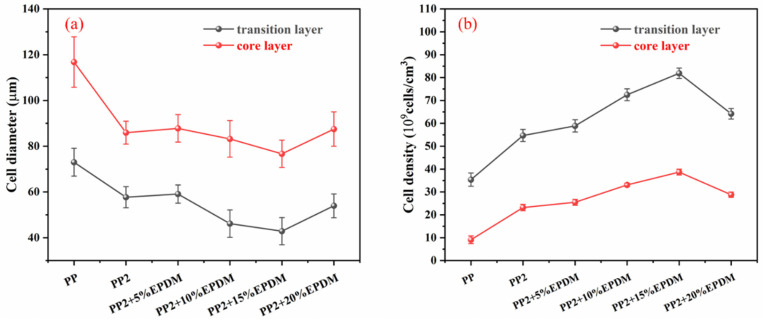
The cell parameters of PP-composite foams perpendicular to the melt-flow direction: (**a**) cell diameter; (**b**) cell density.

**Figure 10 polymers-15-03604-f010:**
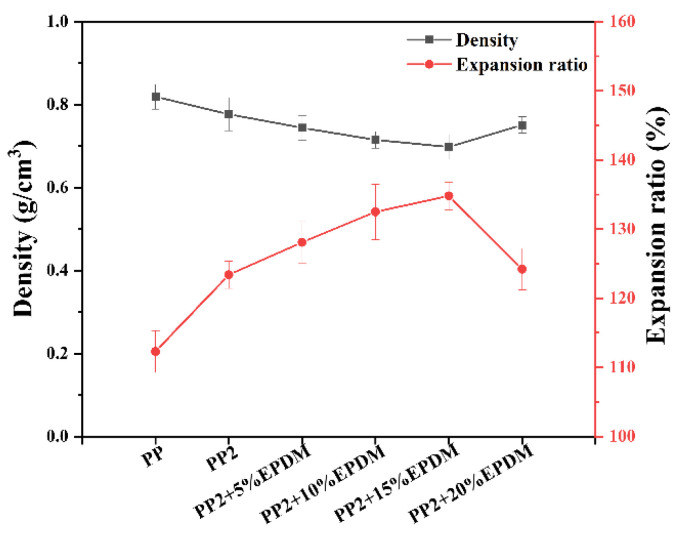
The density and expansion ratio of PP-composite foams.

**Figure 11 polymers-15-03604-f011:**
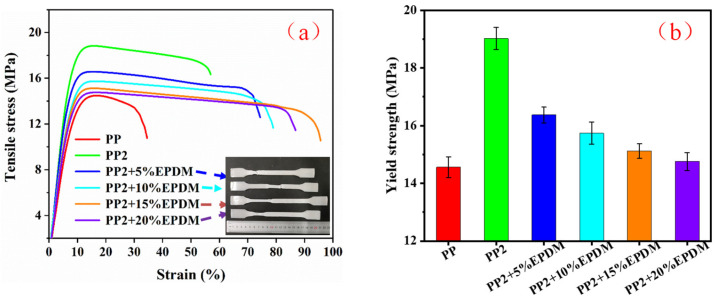
Tensile properties of PP-composite foams: (**a**) tensile stress; (**b**) yield stress.

**Figure 12 polymers-15-03604-f012:**
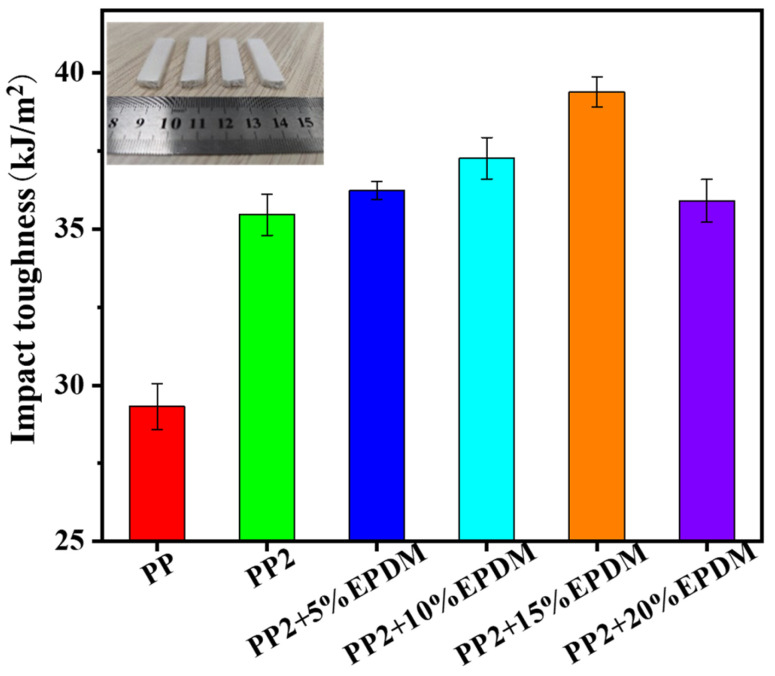
Impact properties of PP-composite foams.

**Figure 13 polymers-15-03604-f013:**
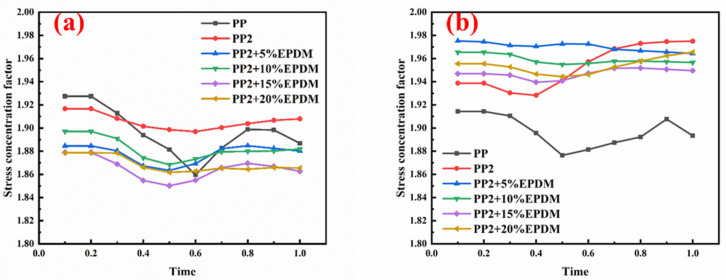
Curves of loading time and stress-concentration factor of the section perpendicular to the melt-flow direction: (**a**) the transition layer; (**b**) the core layer.

**Figure 14 polymers-15-03604-f014:**
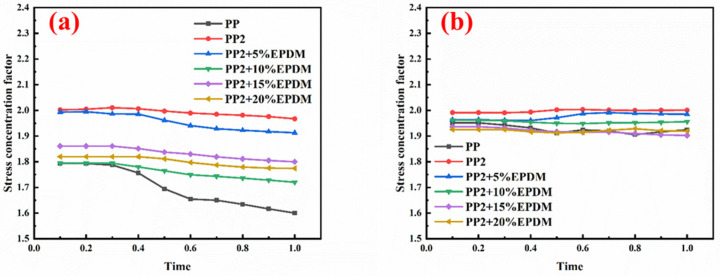
Curves of loading time and stress-concentration factor of the section parallel to the melt-flow direction: (**a**) the transition layer; (**b**) the core layer.

**Table 1 polymers-15-03604-t001:** Materials and parameters.

Materials	Density (g/cm^3^)	Young’s Modulus (MPa)	Poisson’s Ratio
PP	0.92	1350	0.4
EPDM	0.82	6	0.49
nano-CaCO_3_	2.83	10,000	0.31

**Table 2 polymers-15-03604-t002:** The microcellular structure of PP-composite foams perpendicular to the melt-flow direction [25].

Sets	The Core Layer	The Transition Layer
PP	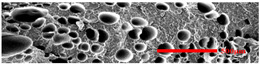	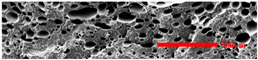
PP2	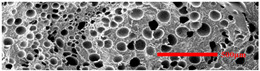	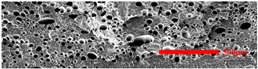
PP2 + 5%EPDM	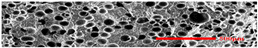	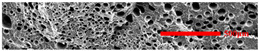
PP2 + 10%EPDM	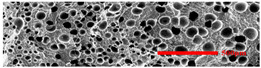	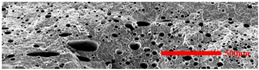
PP2 + 15%EPDM	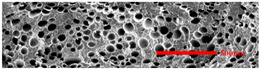	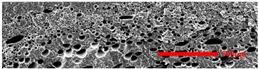
PP2 + 20%EPDM	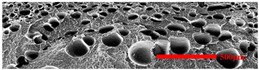	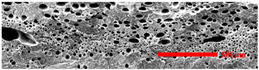

**Table 3 polymers-15-03604-t003:** The microcellular structure of PP-composite foams parallel to the melt-flow direction.

Sets	The Core Layer	The Transition Layer
PP	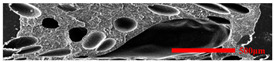	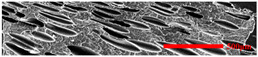
PP2	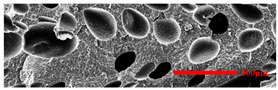	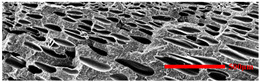
PP2 + 5%EPDM	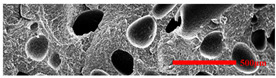	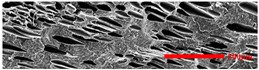
PP2 + 10%EPDM	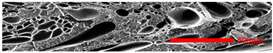	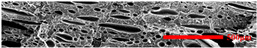
PP2 + 15%EPDM	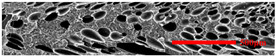	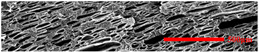
PP2 + 20%EPDM	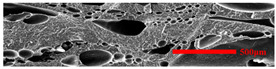	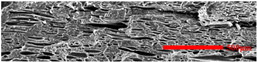

**Table 4 polymers-15-03604-t004:** The stress distribution of the section perpendicular to the melt-flow direction (Time = 1).

Sets	The Core Layer	The Transition Layer
PP	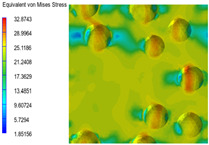	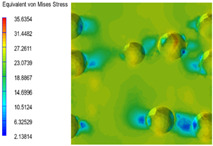
PP2	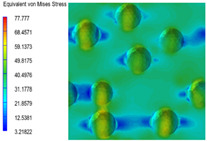	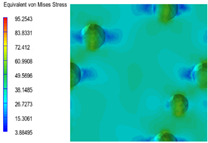
PP2 + 5% EPDM	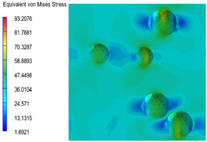	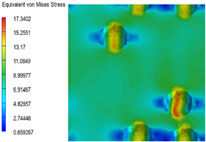
PP2 + 10% EPDM	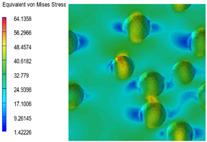	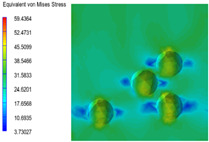
PP2 + 15% EPDM	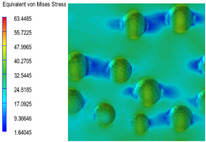	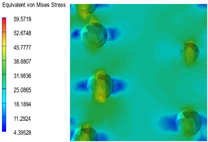
PP2 + 20% EPDM	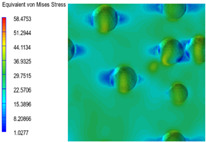	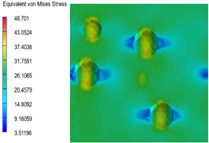

**Table 5 polymers-15-03604-t005:** The stress distribution of the section parallel to the melt-flow direction (Time = 1).

Sets	The Core Layer	The Transition Layer
PP	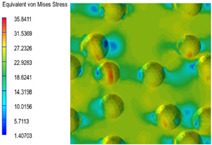	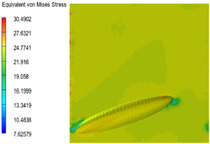
PP2	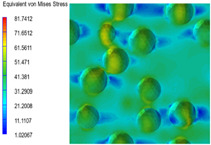	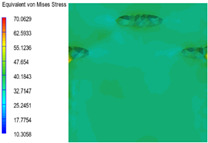
PP2 + 5% EPDM	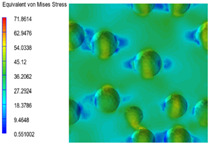	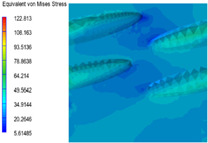
PP2 + 10% EPDM	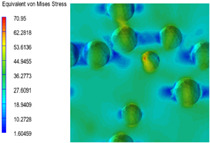	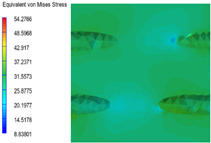
PP2 + 15% EPDM	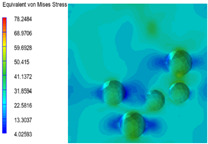	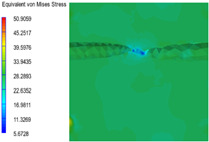
PP2 + 20% EPDM	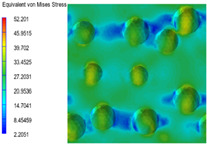	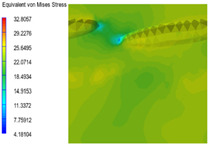

## Data Availability

The raw/processed data required to reproduce these findings cannot be shared at this time, as the data also forms part of an ongoing study.

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
