# Peer review of "The Cellular Structure and Mechanical Properties of Polypropylene/Nano-CaCO3/Ethylene-propylene-diene-monomer Composites Prepared by an In-Mold-Decoration/Microcellular-Injection-Molding Process"

_polymers, 2023, doi:10.3390/polym15173604_

Round 1
Reviewer 1 Report
This is a comprehensive study on three-phase PP composite foam by microcellular injection molding. Simulation is involved on mechanical property analysis of the sophisticated structure. Such a combined experimental and numerical methodology can bring value to the polymeric foam community that people can follow. However, the manuscript is not very well organized and i suggest major revision before publication.
1. the lengthy review on pp composite foams should be simplified with a good focus on those related to this work. avoid touching too much details of other unrelated works.
2. the authors should clarify why they expect the combination of EPDM and CaCO3 could be beneficial. what's the design rationale of this tri-phase structure? focusing on related literatures to supporting the rationale
3. PP/EPDM/talc has been studied before. what's the significance of using CaCO3 instead of talc in this study? what's their hydrophilicity and how it could impact the distribution of the particle?
4.in section 2.4, which specific non-linear model was used for pp? which specific linear model was used for elastomer? how the compounded modulus was calculated? Any hypothesis were made for this calculation?
5. section 3.1, what's the flow regime/shear rate during injection molding? correlation should be made between cell growth and the shear viscosity data in that specific shear rate range
6. section 3.2, why adding elastomer can inhibit the growth of beta crystal?
7. what's the density or expansion ratio of the foamed sample? can density difference impact the tensile strength measurement?
8. from figure8, looks like the EPDM phase is always a discontinuous phase. and it distributed in PP like micro particles. can it really maximize the elasticity of the composite? correct me if I'm wrong.
8. explanation should be given in more detail why adding more EPDM (from 15% to 20%) causes poorer cell morphology. If EPDM are still particle like in the blend, I would assume it will still behave like nucleation agent. In that sense, intuitively, couldn't the cell density be even higher?
9. line 307 to 310, supporting reference should be given for this phenomena
10. looks like figure 12(d) has a higher magnitude than the others subfigures. please double confirm if there is a misuse
there are some gramma error that authors should screen and correct them
Author Response
Thank you very much for your comments and suggestions, we have made serious revisions to our work according to your comments.

Reviewer 2 Report
Review of the manuscript entitled ‘The Cellular Structure and Mechanical Properties of PP/Nano-2 CaCO3/EPDM Composites Prepared by IMD/MIM Process’
Ethylene-Propylene-Diene Monomer and nano-CaCO3 are blended with polypropylene and combined with in-mold decoration to prepare foamed parts by IMD/Microcellular injection molding process. The effects of elastic particles on the cellular structure and mechanical properties of the PP/ Nano-CaCO3 mixture are studied.
The study is interesting and deserves to be published. The manuscript is well written with convincing results of simulations. The SEM images show clearly the structures of the materials. The introduction gives a brief state of the art and allows to the reader to understand the goal of the study.
Author Response
Thank you for your comment, we strive to do much better.
Reviewer 3 Report
The manuscript discusses the fabrication of PP composites using organic (EPDM) and inorganic (CaCO3) as a fillers through microcellular injection molding (MIM) and in-mold decoration (IMD). The obtained material foaming and mechanical properties were studied in detail. The manuscript objective is clear and largely well written.
I would suggest authors to make a detailed discussion on the dispersion of the fillers in the PP matrix. A poor dispersion may also impact the mechanical/foaming properties of the composite.
The evidence for the existence of CaCO3 particles in nano-size is not sufficient as described in Scheme 1. Kindly add sufficient evidence or correct the conveyed information.
Manuscript English is acceptable.
Author Response
Thank you very much for your comments and suggestions, we have made very serious revisions to our work according to your comments

Round 2
Reviewer 1 Report
The manuscript is recommended to be published